# Attentional Pooling for Action Recognition

**Rohit Girdhar**     **Deva Ramanan**
The Robotics Institute, Carnegie Mellon University
http://rohitgirdhar.github.io/AttentionalPoolingAction

## Abstract

We introduce a simple yet surprisingly powerful model to incorporate attention in action recognition and human object interaction tasks. Our proposed attention module can be trained with or without extra supervision, and gives a sizable boost in accuracy while keeping the network size and computational cost nearly the same. It leads to significant improvements over state of the art base architecture on three standard action recognition benchmarks across still images and videos, and establishes new state of the art on MPII dataset with 12.5% relative improvement. We also perform an extensive analysis of our attention module both empirically and analytically. In terms of the latter, we introduce a novel derivation of bottom-up and top-down attention as low-rank approximations of bilinear pooling methods (typically used for fine-grained classification). From this perspective, our attention formulation suggests a novel characterization of action recognition as a fine-grained recognition problem.

## 1   Introduction

Human action recognition is a fundamental and well studied problem in computer vision. Traditional approaches to action recognition relied on object detection [11, 19, 57], articulated pose [29, 34, 35, 55, 57], dense trajectories [52, 53] and part-based/structured models [9, 56, 58]. However, more recently these methods have been surpassed by deep CNN-based representations [18, 30, 42, 47]. Interestingly, even video based action recognition has benefited greatly from advancements in image-based CNN models [20, 22, 43, 46]. With the exception of a few 3D-conv-based methods [33, 47, 49], most approaches [12, 14, 15, 17, 54], including the current state of the art [54], use a variant of discriminatively trained 2D-CNN [22] over the appearance (frames) and in some cases motion (optical flow) modalities of the input video.

**Attention:** While using standard deep networks over the full image have shown great promise for the task [54], it raises the question of whether action recognition can be considered as a general classification problem. Some recent works have tried to generate more *fine-grained* representations by extracting features around human pose keypoints [8] or on object/person bounding boxes [18, 30]. This form of 'hard-coded attention' helps improve performance, but requires labeling (or detecting) objects or human pose. Moreover, these methods assume that focusing on the human or its parts is always useful for discriminating actions. This might not necessarily be true for all actions; some actions might be easier to distinguish using the background and context, like a 'basketball shoot' vs a 'throw'; while others might require paying close attention to objects being interacted by the human, like in case of 'drinking from mug' vs 'drinking from water bottle'.

**Our work:** In this work, we propose a simple yet surprisingly powerful network modification that learns attention maps which focus computation on specific parts of the input relevant to the task at hand. Our attention maps can be learned without any additional supervision and automatically lead to significant improvements over the baseline architecture. Our formulation is simple-to-implement, and can be seen as a natural extension of average pooling into a "weighted" average pooling with image-specific weights. Our formulation also provides a novel factorization of attentional processing

into bottom-up saliency combined with top-down attention. We further experiment with adding human pose as an intermediate supervision to train our attention module, which encourages the network to look for human object interactions. While this makes little difference to the performance of image-based recognition models, it leads to a larger improvement on video datasets as videos consist of large number of 'non-iconic' frames where the subject of object of actions may not be at the center of focus.

**Our contributions:** (1) An easy-to-use extension of state-of-the-art base architectures that incorporates attention to give significant improvement in action recognition performance at virtually negligible increase in computation cost; (2) Extensive analysis of its performance on three action recognition datasets across still images and videos, obtaining state of the art on MPII and HMDB-51 (RGB, single-frame models) and competitive on HICO; (3) Analysis of different base architectures for applicability of our attention module; and (4) Mathematical analysis of our proposed attention module and showing its equivalence to a rank-1 approximation of second order or bilinear pooling (typically used in fine grained recognition methods [16, 26, 28]) suggesting a novel characterization of action recognition as a fine grained recognition problem.

## 2 Related Work

Human action recognition is a well studied problem with various standard benchmarks spanning across still images [7, 13, 34, 36, 58] and videos [24, 27, 41, 45]. The newer image based datasets such as HICO [7] and MPII [34] are large and highly diverse, containing 600 and 393 classes respectively. In contrast, collecting such diverse video based action datasets is hard, and hence existing popular benchmarks like UCF101 [45] or HMDB51 [27] contain only 101 and 51 categories each. This in turn has lead to much higher baseline performance on videos, eg. $\sim 94\%$ [54] classification accuracy on UCF101, compared to images, eg. $\sim 32\%$ [30] mean average precision (mAP) on MPII.

**Features:** Video based action recognition methods focus on two main problems: action classification and (spatio-)temporal detection. While image based recognition problems, including action recognition, have seen a large boost with the recent advancements in deep learning (e.g., MPII performance went up from 5% mAP [34] to 27% mAP [18]), video based recognition still relies on hand crafted features such as iDT [53] to obtain competitive performance. These features are computed by extracting appearance and motion features along densely sampled point trajectories in the video, aggregated into a fixed length representation by using fisher vectors [32]. Convolutional neural network (CNN) based approaches to video action recognition have broadly followed two main paradigms: (1) Multi-stream networks [42, 54] which split the input video into multiple modalities such as RGB, optical flow, warped flow etc, train standard image based CNNs on top of those, and late-fuse the predictions from each of the CNNs; and (2) 3D Conv Networks [47, 49] which represent the video as a spatio-temporal blob and train a 3D convolutional model for action prediction. In terms of performance, 3D conv based methods have been harder to scale and multi-stream methods [54] currently hold state of the art performance on standard benchmarks. Our approach is complementary to these paradigms and the attention module can be applied on top of either. We show results on improving action classification over state of the art multi-stream model [54] in experiments.

**Pose:** There have also been previous works in incorporating human pose into action recognition [8, 10, 60]. In particular, P-CNN [8] computes local appearance and motion features along the pose keypoints and aggregates those over the video for action prediction, but is not end-to-end trainable. More recent work [60] adds pose as an additional stream in chained multi-stream fashion and shows significant improvements. Our approach is complementary to these approaches as we use pose as a regularizer in learning spatial attention maps to weight regions of the RGB frame. Moreover, our method is not constrained by pose labels, and as we show in experiments, can show effective performance with pose predicted by existing methods [4] or even without using pose.

**Hard attention:** Previous works in image based action recognition have shown impressive performance by incorporating evidence from the human, context and pose keypoint bounding boxes [8, 18, 30]. Gkioxari *el al.* [18] modified R-CNN pipeline to propose R*CNN, where they choose an auxiliary box to encode context apart from the human bounding box. Mallya and Lazebnik [30] improve upon it by using the full image as the context and using multiple instance learning (MIL) to reason over all humans present in the image to predict an action label for the image. Our approach gets rid of the bounding box detection step and improves over both these methods by automatically learning to attend to the most informative parts of the image for the task.

**Soft attention:** There has been relatively little work that explores unconstrained 'soft' attention for action recognition, with the exception of [39, 44] for spatio-temporal and [40] for temporal attention. Importantly, all these consider a video setting, where a LSTM network predicts a spatial attention map for the current frame. Our method, however, uses a single frame to both predict and apply spatial attention, making it amenable to both single image and video based use cases. [44] also uses pose keypoints labeled in 3D videos to drive attention to parts of the body. In contrast, we learn an unconstrained attention model that frequently learns to look around the human body for objects that make it easier to classify the action.

**Second-order pooling:** Because our model uses a single set of appearance features to both predict and apply an attention map, this makes the output *quadratic* in the features (Sec. 3.1). This observation allows us to implement attention through second-order or bilinear pooling operations [28], made efficient through low-rank approximations [16, 25, 26]. Our work is most related to [26], who point out when efficiently implemented, low-rank approximations avoid explicitly computing second-order features. We point out that a rank-1 approximation of second-order features is equivalent to an attentional model sometimes denoted as "self attention" [50]. Exposing this connection allows us to explore several extensions, including variations of bottom-up and top-down attention, as well as regularized attention maps that make use of additional supervised pose labels.

## 3   Approach

Our attentional pooling module is a trainable layer that plugs in as a replacement for a pooling operation in any standard CNN. As most contemporary architectures [20, 22, 46] are fully convolutional with an average pooling operation at the end, our module can be used to replace that operation with an attention-weighted pooling. We now derive the pooling layer as an efficient low-rank approximation to second order pooling (Sec. 3.1). Then, we describe our network architecture that incorporates this attention module and explore a pose-regularized variant of the same (Sec. 3.2).

### 3.1   Attentional pooling as low-rank approximation of second-order pooling

Let us write the layer to be pooled as $X \in R^{n \times f}$, where $n$ is the number of spatial locations (e.g., $n = 16 \times 16 = 256$) and $f$ is the number of channels (e.g., 2048). Standard sum (or max) pooling would reduce this to vector in $R^{f \times 1}$, which could then be processed by a "fully-connected" weight vector $\mathbf{w} \in R^{f \times 1}$ to generate a classification score. We will denote matrices with upper case letters, and vectors with lower-case bold letters. For the moment, assume we are training a binary classifier (we generalize to more classes later in the derivation). We can formalize this pipeline with the following notation:

$$score_{pool}(X) = \mathbf{1}^T X \mathbf{w}, \qquad \text{where} \qquad X \in R^{n \times f}, \mathbf{1} \in R^{n \times 1}, \mathbf{w} \in R^{f \times 1} \qquad (1)$$

where $\mathbf{1}$ is a vector of all ones and $\mathbf{x} = \mathbf{1}^T X \in R^{1 \times f}$ is the (transposed) sum-pooled feature.

**Second-order pooling:** Following past work on second-order pooling [5], let us construct the feature $X^T X \in R^{f \times f}$. Prior work has demonstrated that such second-order statistics can be useful for fine-grained classification [28]. Typically, one then "vectorizes" this feature, and learns a $f^2$ vector of weights to generate a score. If we write the vector of weights as a $f \times f$ matrix, the inner product between the two vectorized quantities can be succinctly written using the trace operator[1]. The key identity, $Tr(AB^T) = dot(A(:), B(:))$ (using matlab notation), can easily be verified by plugging in the definition of a trace operator. This allows us to write the classification score as follows:

$$score_{order2}(X) = Tr(X^T X W^T), \qquad \text{where} \qquad X \in R^{n \times f}, W \in R^{f \times f} \qquad (2)$$

**Low-rank second-order pooling:** Let us approximate matrix $W$ with a rank-1 approximation, $W = \mathbf{a}\mathbf{b}^T$ where $\mathbf{a}, \mathbf{b} \in R^{f \times 1}$. Plugging this into the above yields a novel formulation of attentional

pooling:

$$score_{attention}(X) = Tr(X^T X \mathbf{b} \mathbf{a}^T), \qquad \text{where} \qquad X \in R^{n \times f}, \mathbf{a}, \mathbf{b} \in R^{f \times 1} \qquad (3)$$

$$= Tr(\mathbf{a}^T X^T X \mathbf{b}) \qquad (4)$$

$$= \mathbf{a}^T X^T X \mathbf{b} \qquad (5)$$

$$= \mathbf{a}^T \left( X^T (X \mathbf{b}) \right) \qquad (6)$$

where (4) makes use of the trace identity that $Tr(ABC) = Tr(CAB)$ and (5) uses the fact that the trace of a scalar is simply the scalar. The last line (6) gives efficient implementation of attentional pooling: given a feature map $X$, compute an attention map over all $n$ spatial locations with $\mathbf{h} = X\mathbf{b} \in R^{n \times 1}$, that is then used to compute a weighted average of features $\mathbf{x} = X^T \mathbf{h} \in R^{f \times 1}$. This weighted-average feature is then pushed through a linear model $\mathbf{a}^T \mathbf{x}$ to produce the final score.

Interestingly, (6) can also be written as the following:

$$score_{attention}(X) = \left( (X\mathbf{a})^T X \right) \mathbf{b} \qquad (7)$$

$$= (X\mathbf{a})^T (X\mathbf{b}) \qquad (8)$$

The first line illustrates that the attentional heatmap can also be seen as $X\mathbf{a} \in R^{n \times 1}$, with $\mathbf{b}$ being the classifier of the attentionally-pooled feature. The second line illustrates that our formulation is in fact symmetric, where the final score can be seen as the inner product between *two* attentional heatmaps defined over all $n$ spatial locations. Fig. 1a illustrates our approach.

**Top-down attention:** To generate prediction for multiple classes, we replace the weight matrix from (2) with class-specific weights:

$$score_{order2}(X, k) = Tr(X^T X W_k^T), \qquad \text{where} \qquad X \in R^{n \times f}, W_k \in R^{f \times f} \qquad (9)$$

One could apply a similar derivation to produce class-specific vectors $\mathbf{a}_k$ and $\mathbf{b}_k$, each of them generating a class-specific attention map. Instead, we choose to distinctly model class-specific "top-down" attention [3, 48, 59] from bottom-up visual saliency that is class-agnostic [37]. We do so by forcing one of the attention parameter vectors to be class-agnostic - e.g., $b_k = b$. This makes our final low-rank attentional model

$$score_{attention}(X, k) = \mathbf{t}_k^T \mathbf{h}, \qquad \text{where} \qquad \mathbf{t}_k = X\mathbf{a}_k, \mathbf{h} = X\mathbf{b} \qquad (10)$$

equivalent to an inner product between top-down (class-specific) $\mathbf{t}_k$ and bottom-up (saliency-based) $\mathbf{h}$ attention maps. Our approach of combining top-down and botom-up attentional maps is reminiscent of biologically-motivated schemes that *modulate* saliency maps with top-down cues [31]. This suggests that our attentional model can also be implemented using a single, combined attention map defined over all $n$ spatial locations:

$$score_{attention}(X, k) = \mathbf{1}^T \mathbf{c}_k, \qquad \text{where} \qquad \mathbf{c}_k = \mathbf{t}_k \circ \mathbf{h}, \qquad (11)$$

where $\circ$ denotes element-wise multiplication and $\mathbf{1}$ is defined as before. We visualize the combined, top-down, and bottom-up attention maps $\mathbf{c}_k, \mathbf{t}_k, \mathbf{h} \in R^{n \times 1}$ in our experimental results.

**Average pooling (revisited):** The above derivation allows us to revisit our average pooling formulation from (1), replacing weights $\mathbf{w}$ with class-specific weights $\mathbf{w}_k$ as follows:

$$score_{top-down}(X, k) = \mathbf{1}^T X \mathbf{w}_k = \mathbf{1}^T \mathbf{t}_k \quad \text{where} \quad \mathbf{t}_k = X\mathbf{w}_k \qquad (12)$$

From this perspective, the above derivation gives the ability to generate top-down attentional maps from *existing* average-pooling networks. While similar observations have been pointed out before [59], it naturally emerges as a special case of our bottom-up and top-down formulation of attention.

## 3.2 Network Architecture

We now describe our network architecture to implement the attentional pooling described above. We start from a state of the art base architecture, ResNet-101 [20]. It consists of a stack of 'modules', each of which contains multiple convolutional, pooling or identity mapping streams. It finally generates a $n_1 \times n_2 \times f$ spatial feature map, which is average pooled to get a $f$-dimensional vector and is then classified using a linear classifier.

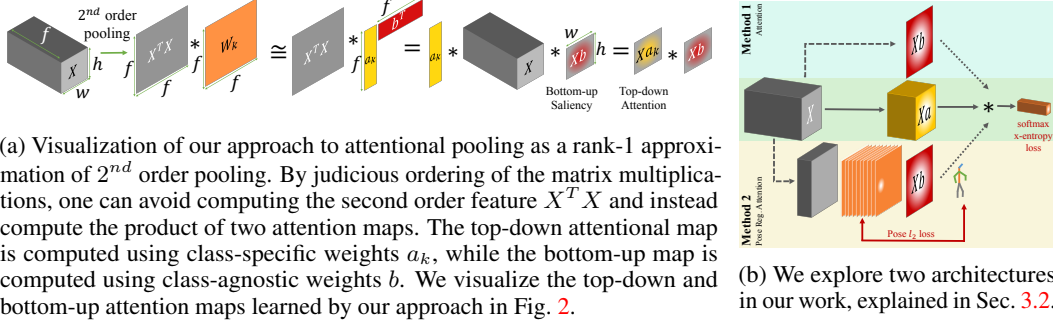

(a) Visualization of our approach to attentional pooling as a rank-1 approximation of $2^{nd}$ order pooling. By judicious ordering of the matrix multiplications, one can avoid computing the second order feature $X^T X$ and instead compute the product of two attention maps. The top-down attentional map is computed using class-specific weights $a_k$, while the bottom-up map is computed using class-agnostic weights $b$. We visualize the top-down and bottom-up attention maps learned by our approach in Fig. 2.

(b) We explore two architectures in our work, explained in Sec. 3.2.

Figure 1: Visualization of our derivation and final network architectures.

Our attention module plugs in at the last layer, after the spatial feature map. As shown in Fig. 1b (Method 1), we predict a single channel bottom-up saliency map of same spatial resolution as the last feature map, using a linear classifier on top of it ($X\mathbf{b}$). Similarly, we also generate the $n_1 \times n_2 \times K$ dimensional top-down attention map $X\mathbf{a}$, where $K$ is number of classes. The two attention maps are multiplied and spatially averaged to generate the $K$-dimensional output predictions ($(X\mathbf{a})^T(X\mathbf{b})$). These operations are equivalent to first multiplying the features with saliency ($X^T(X\mathbf{b})$) and then passing through a classifier ($\mathbf{a}(X^T(X\mathbf{b}))$).

**Pose:** While this unconstrained attention module automatically learns to focus on relevant parts and gives a sizable boost in accuracy, we take inspiration from previous work [8] and use human pose keypoints to guide the attention. As shown in Fig. 1b (Method 2), we use a two-layer MLP on top of the last layer to predict a 17 channel heatmap. The first 16 channels correspond to human pose keypoints and incur a $l_2$ loss against labeled (or detected, using [4]) pose) The final channel is used as an unconstrained bottom-up attention map, as before. We refer to this method as pose-regularized attention, and it can be thought of as a non-linear extension of previous attention map.

## 4 Experiments

**Datasets:** We experiment with three recent, large scale action recognition datasets, across still images and videos, namely MPII, HICO and HMDB51. **MPII Human Pose Dataset** [34] contains 15205 images labeled with up to 16 human body keypoints, and classified into one of 393 action classes. It is split into train, val (from authors of [18]) and test sets, with 8218, 6987 and 5708 images each. We use the val set to compare with [18] and for ablative analysis while the final test results are obtained by emailing our results to authors of [34]. The dataset is highly imbalanced and the evaluation is performed using mean average precision (mAP) to equally weight all classes. **HICO** [7] is a recently introduced dataset with labels for 600 human object interactions (HOI) combining 117 actions with 80 objects. It contains 38116 training and 9658 test images, with each image labeled with all the HOIs active for that image (multi-label setting). Like MPII, this dataset is also highly unbalanced and evaluation is performed using mAP over classes. Finally, to verify our method's applicability to video based action recognition, we experiment with a challenging trimmed action classification dataset, **HMDB51** [27]. It contains 6766 realistic and varied video clips from 51 action classes. Evaluation is performed using average classification accuracy over three train/test splits from [23], each with 3570 train and 1530 test videos.

**Baselines:** Throughout the following sections, we compare our approach first to the standard base architecture, mostly ResNet-101 [20], without the attention-weighted pooling. Then we compare to other reported methods and previous state of the art on the respective datasets.

**MPII:** We train our models for 393-way action classification on MPII with softmax cross-entropy loss for both the baseline ResNet and our attentional model. We compare our performance in Tab. 1. Our unconstrained attention model clearly out-performs the base ResNet model, as well as previous state of the art methods involving detection of multiple contextual bounding boxes [18] and fusion of full image with human bounding box features [30]. Our pose-regularized model performs best, though the improvement is small. We visualize the attention maps learned in Fig. 2.

**HICO:** We train our model on HICO similar to MPII, and compare our performance in Tab. 2. Again, we see a significant 5% boost over our base ResNet model. Moreover, we out-perform all

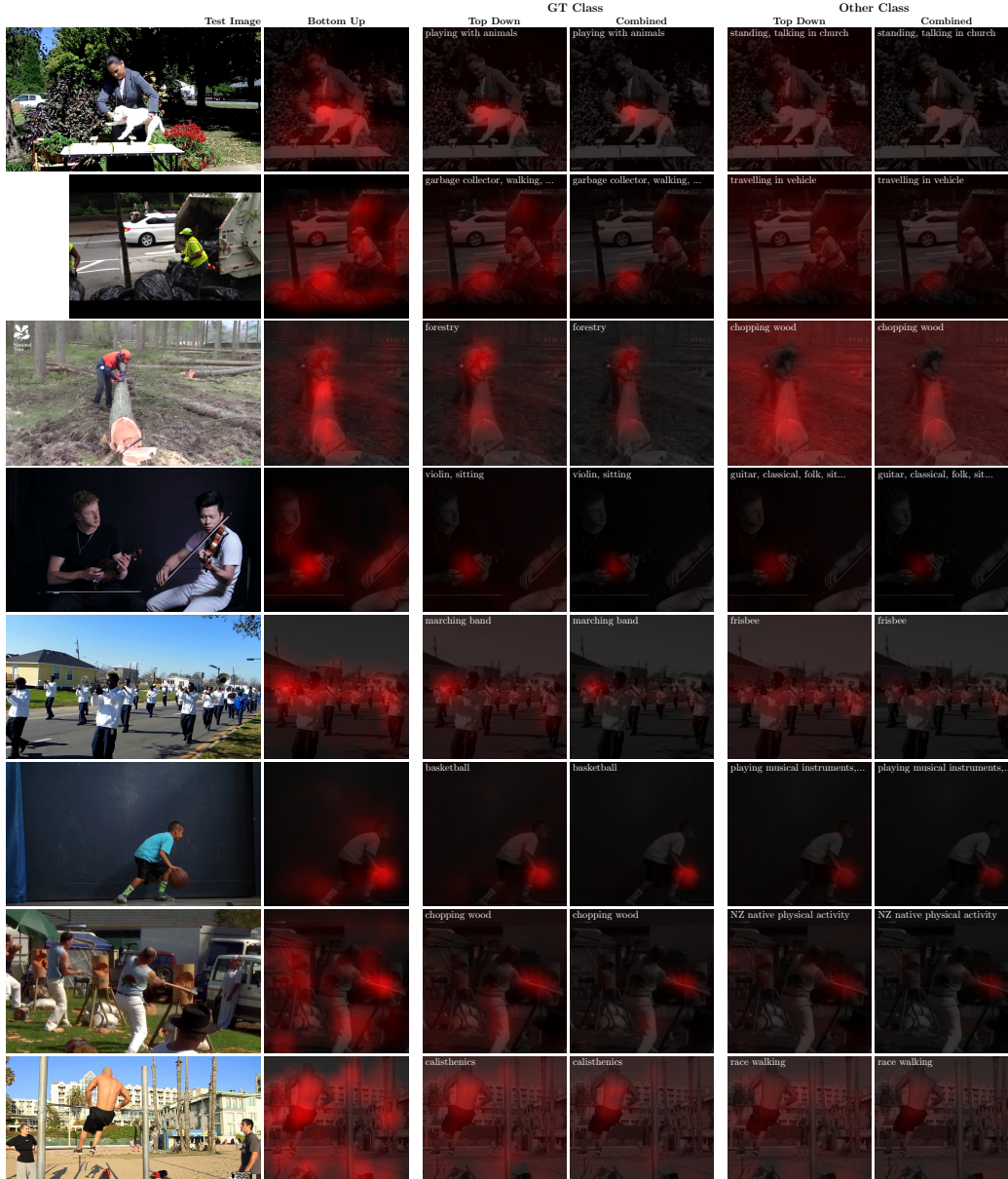

Figure 2: Auto-generated (not hand-picked) visualization of bottom-up $(X\mathbf{b})$, top-down $(X\mathbf{a}_k)$ and combined $((X\mathbf{a}_k) \circ (X\mathbf{b}))$ attention on validation images in MPII, that see largest improvement in softmax score for correct class when trained with attention. Since the top-down/combined maps are class specific, we mention the class name for which they are generated for on top left of those heatmaps. We consider 2 classes, the ground truth (GT) for the image, and the class on which it gets lowest softmax score. The attention maps for GT class focus on the objects most useful for distinguishing the class. Though the top-down and combined maps look similar in many cases, they do capture different information. For example, for a garbage collector action (second row), top-down also focuses on the vehicles in background, while the combined map narrows focus down to the garbage bags. (Best viewed zoomed-in on screen)

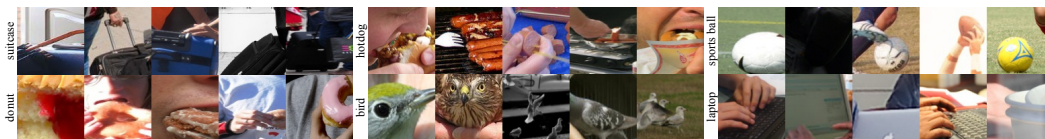

Figure 3: We crop a 100px patch around the attention peak for all images containing an HOI involving a given object, and show 5 **randomly picked** patches for 6 object classes here. This suggests our attention model learns to look for objects to improve HOI detection.

Table 1: Action classification performance on MPII dataset. Validation (Val) performance is reported on train set split shared by authors of [18]. Test performance obtained from training on complete train set and submitting our output file to authors of [34]. Note that even though our pose regularized model uses pose labels at training time for regularizing attention, it does not require any pose input at test time. The **top**-half corresponds to a diagnostic analysis of our approach with different base networks. Attention provides a strong 4% improvement for baseline networks with larger spatial resolution (e.g., ResNet). Please see text for additional discussion. The **bottom**-half reports prior work that makes use of object bounding boxes/pose. Our method performs slightly better with pose annotations (on training data), but even without *any* pose or detection annotations, we outperform all prior work.

| Method | Full Img | Bbox | Pose | MIL | Val (mAP) | Test (mAP) |
|---|---|---|---|---|---|---|
| Inception-V2 (ours) | ✓ | | | | 25.2 | - |
| ResNet101 (ours) | ✓ | | | | 26.2 | - |
| Attn. Pool. (I-V2) (ours) | ✓ | | | | 24.3 | - |
| Attn. Pool. (R-101) (ours) | ✓ | | | | **30.3** | **36.0** |
| Dense Trajectory + Pose [34] | ✓ | | ✓ | | - | 5.5 |
| VGG16, RCNN [18] | | ✓ | | | 16.5 | - |
| VGG16, R*CNN [18] | | ✓ | | | 21.7 | 26.7 |
| VGG16, Fusion (best) [30] | ✓ | ✓ | | | - | 32.2 |
| VGG16, Fusion+MIL (best) [30] | ✓ | ✓ | | ✓ | - | 31.9 |
| Pose Reg. Attn. Pooling (R-101) (ours) | ✓ | | ✓ | | **30.6** | **36.1** |

Table 2: Multi-label HOI classification performance on HICO dataset. The **top**-half compares our performance to other full image-based methods. The **bottom**-half reports methods that use object bounding boxes/pose. Our model out-performs various approaches that need bounding boxes, multi-instance learning (MIL) or specialized losses, and achieves performance competitive to state of the art. Note that even though our pose regularized model uses computed pose labels at training time, it does not require any pose input at test time.

| Method | Full Im. | Bbox/Pose | MIL | Wtd Loss | mAP |
|---|---|---|---|---|---|
| AlexNet+SVM [7] | ✓ | | | | 19.4 |
| VGG16, full image [30] | ✓ | | | | 29.4 |
| ResNet101, full image (ours) | ✓ | | | | 30.2 |
| ResNet101 with CBP [16] (impl. from [1]) | ✓ | | | | 26.8 |
| Attentional Pooling (R-101) (ours) | ✓ | | | | **35.0** |
| R*CNN [18] (reported in [30]) | | ✓ | ✓ | | 28.5 |
| Scene-RCNN [18] (reported in [30]) | ✓ | ✓ | ✓ | | 29.0 |
| Fusion (best reported) [30] | ✓ | ✓ | ✓ | | 33.8 |
| Pose Regularized Attentional Pooling (R101) (ours) | ✓ | ✓ | | | **34.6** |
| Fusion, weighted loss (best reported) [30] | ✓ | ✓ | ✓ | ✓ | 36.1 |

previous methods, including ones that use detection bounding boxes at test time except one [30], when that is trained with a specialized weighted loss for this dataset. It is also worth noting that the full image-only performance of VGG and ResNet were comparable in our experiments (29.4% and 30.2%), suggesting that our approach shows larger relative improvement over a similar starting baseline. Though we did not experiment with the same optimization setting as [30], we believe it will give similar improvements there as well. Since this dataset also comes with labels decomposed into actions and objects, we visualize what our attention model looks for, given images containing interactions with a specific object. As Fig. 3 shows, the attention peak is typically close to the object of interest, showing the importance of detecting objects in HOI detection tasks. Moreover, this suggests that our attention maps can also function as weak-supervision for object detection.

**HMDB51:** Next, we apply our attentional method to the RGB stream of the current state of the art single-frame deep model on this dataset, TSN [54]. TSN extends the standard two-stream [42] architecture by using a much deeper base architecture [22] along with enforcing consensus over multiple frames from the video at training time. For the purpose of this work, we focus on the RGB stream only but our method is applicable to flow/warped-flow streams as well. We first train a TSN model using ResNet-101 as base architecture after re-sizing input frames to 450px. This ensures larger spatial dimensions of the output ($14 \times 14$), hence ensuring the last-layer features are amenable to attention. Though our base ResNet model does worse than BN-inception TSN model, as Tab. 3 shows, using our attention module improves the base model to do comparably well. Interestingly, on this dataset regularizing the attention through pose gives a significant boost in

Table 3: Action classification performance on HMDB51 dataset using only the RGB stream of a two-stream model. Our base ResNet stream training is done over 480px rescaled images, same as used in our attention model for comparison purposes. Our pose based attention model out-performs the base network by large margin, as well as the previous RGB stream (single-frame) state-of-the-art, TSN [54].

| Method | Split 1 | Split 2 | Split 3 | Avg |
|---|---|---|---|---|
| TSN, BN-inception (RGB) [54] (Via email with authors) | **54.4** | 49.5 | 49.2 | 51.0 |
| ActionVLAD [17] | 51.2 | - | - | 49.8 |
| RGB Stream, ResNet50 (RGB) [14] (reported at [2]) | - | - | - | 48.9 |
| RGB Stream, ResNet152 (RGB) [14] (reported at [2]) | - | - | - | 46.7 |
| TSN, ResNet101 (RGB) (ours) | 48.2 | 46.5 | 46.7 | 47.1 |
| Linear Attentional Pooling (ours) | 51.1 | **51.6** | 49.7 | 50.8 |
| Pose regularized Attentional Pooling (ours) | **54.4** | 51.1 | **50.9** | **52.2** |

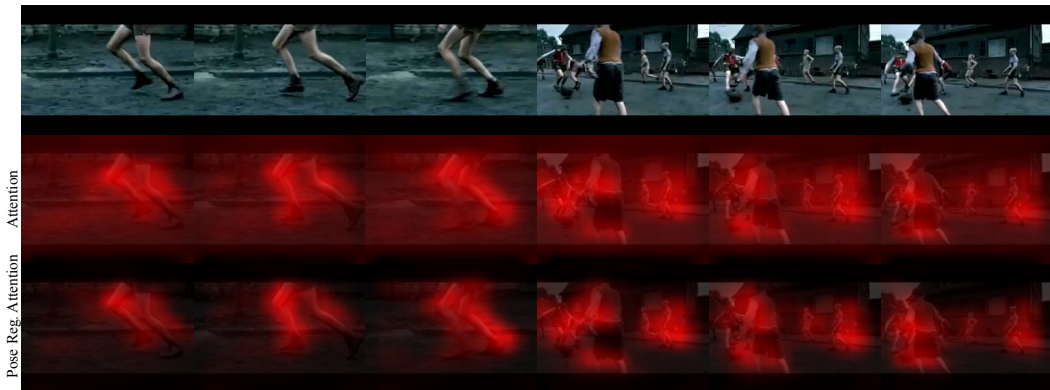

Figure 4: Attention maps with linear attention and pose regularized attention on a video from HMDB. Note the pose-guided attention is better able to focus on regions of interest in the non-iconic frames.

performance, out-performing TSN and establishing new state of the art on the RGB-only single-frame model for HMDB. We visualize the attention maps with normal and pose-regularized attention in Fig. 4. The pose regularized attention are more peaky near the human than their linear counterparts. This potentially explains the improvement using pose on HMDB while it does not help as much on HICO or MPII; HICO and MPII, being image based datasets typically have 'iconic' images, with the subjects and objects of action typically in the center and focus of the image. Video frames in HMDB, on the other hand, may have the subject move all across the frame throughout the video, and hence additional supervision through pose at training time helps focus the attention at the right spot.

**Full-rank pooling:** Given our formulation of attention as low-rank second-order pooling, a natural question is what would be the performance of a full-rank model? Explicitly computing the second-order features of size $f \times f$ for $f = 2048$ (and learning the associated classifier) is cumbersome. Instead, we make use of the compact bilinear approach (CBP) of [16], which generates a low-dimensional approximation of full bilinear pooling [28] using the TensorSketch algorithm. To keep the final output comparable to our attentional-pooled model, we project to $f = 2048$ dimensions. We find it performs slightly *worse* than simple average pooling in Table 2. Note that we use an existing implementation [1] with minimal hyper-parameter optimization, and leave a more rigorous comparison to future work.

**Rank-$P$ approximation:** While a full-rank model is cumbersome, we can still explore the effect of using a higher, $P$-rank approximation. Essentially, a rank-$P$ approximation generates $P$ (1-channel) bottom-up and ($C$ channel) top-down attention maps, and the final prediction is the product of corresponding heatmaps, summed over $P$. On MPII, we obtain mAP of 30.3, 29.9, 30.0 for $P$=1, 2 and 5 respectively, showing that the validation performance is relatively stable with $P$. We do observe a drop in training loss with a higher $P$, indicating that a higher-rank approximation could be useful for harder datasets and tasks.

**Per-class attention maps:** As we described in Sec. 3.1, our inspiration for combining class-specific and class-agnostic classifiers (i.e. top-down and bottom-up attention respectively), came from the Neuroscience literature on integrating top-down and bottom-up attention [31]. However, our model

can also be extended to learn completely class-specific attention maps, by predicting $C$ bottom-up attention maps, and combining each map with the corresponding softmax classifier for that class. We experiment with this idea on MPII and obtain a mAP of 27.9 with 393 (=num-classes) attention maps, compared to 30.3% with 1 map, and 26.2% without attention. On further analysis we observe that both models achieve near perfect mAP on training data, implying that adding more parameters with multiple attention maps leads to over-fitting on the relatively small MPII trainset. However, this may be a viable approach for larger datasets.

**Diagnostics:** It is natural to consider variants of our model that only consider the bottom-up or top-down attentional map. As derived in (12), baseline models with average pooling are equivalent to "top-down-only" attention models, which are resoundingly outperformed by our joint bottom-up and top-down model. It is not clear how to construct a bottom-up only model, since it is class-agnostic, making it difficult to produce class-specific scores. Rather, a reasonable approximation might be applying an off-the-shelf (bottom-up) saliency method used to limit the spatial region that features are averaged over. Our initial experiments with existing saliency-based methods [21] were not promising.

**Base Network:** Finally, we analyze the choice of base architecture for the effectiveness of our proposed attentional pooling module. In Tab. 1, we compare the improvement using attention over ResNet-101 (R-101) [20] and an BN-Inception (I-V2) [22]. Both models perform comparably when trained for full image, however, while we see a 4% improvement on R-101 on using attention, we do not see similar improvements for I-V2. This points to an important distinction in the two architectures, i.e., Inception-style models are designed to be faster in inference and training by rapidly down sampling input images in initial layers through max-pooling. While this reduces the computational cost for later layers, it leads to most layers having very large receptive fields, and hence later neurons have effective access to all of the image pixels. This suggests that all the spatial features at the last layer could be highly similar. In contrast, R-101 downscales the spatial resolution gradually, allowing the last layer features to specialize to different parts of the image, hence benefiting more from attentional pooling. This effect was further corroborated by our experiments on HMDB, where using the standard 224px input resolution showed no improvement with attention, while the same image resized to 450px at input time did. This initial resize ensures the last-layer features are sufficiently distinct to benefit from attentional pooling.

## 5 Discussion and Conclusion

An important distinction of our model from some previous works [18, 30] is that it does not explicitly model action at an instance or bounding-box level. This, in fact, is a strength of our model; making it capable of attending to objects outside of any person-instance bounding box (such as bags of garbage for "garbage collecting", in Fig 2). In theory, our model can also be applied to instance-level action recognition by applying attentional pooling over an instance's RoI features. Such a model would learn to look at different parts of human body and its interactions with nearby objects. However, it's notable that most existing action datasets, including [6, 7, 27, 34, 41, 45], come with only frame or video level labels; and though [18, 30] are designed for instance-level recognition, they are not applied as such. They either copy image level labels to instances or use multiple-instance learning, either of which can be used in conjunction with our model. Another interesting connection that emerges from our work is the relation between second-order pooling and attention. The two communities are traditionally seen as distinct, and our work strongly suggests that they should mix: as newer action datasets become more fine-grained, we should explore second-order pooling techniques for action recognition. Similarly, second-order pooling can serve as a simple but strong baseline for the attention community, which tends to focus on more complex sequential attention networks (based on RNNs or LSTMs). It is also worth noting that similar ideas involving self attention and bilinear models have recently also shown significant improvements in other tasks like image classification [51], language translation [50] and visual question answering [38].

**Conclusion:** We have introduced a simple formulation of attention as low-rank second-order pooling, and illustrate it on the task of action classification from single (RGB) images. Our formulation allows for explicit integration of bottom-up saliency and top-down attention, and can take advantage of additional supervision when needed (through pose labels). Our model produces competitive or state-of-the-art results on widely benchmarked datasets, by learning where to look when pooling features across an image. Finally, it is easy to implement and requires few additional parameters, making it an attractive alternative to standard pooling, which is a ubiquitous operation in nearly all contemporary deep networks.

**Acknowledgements:** Authors would like to thank Olga Russakovsky for initial review. This research was supported in part by the National Science Foundation (NSF) under grant numbers CNS-1518865 and IIS-1618903, and the Defense Advanced Research Projects Agency (DARPA) under Contract No. HR001117C0051. Additional support was provided by the Intel Science and Technology Center for Visual Cloud Systems (ISTC-VCS). Any opinions, findings, conclusions or recommendations expressed in this material are those of the authors and do not necessarily reflect the view(s) of their employers or the above-mentioned funding sources.

## Footnotes

[1] https://en.wikipedia.org/wiki/Trace_(linear_algebra)

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
