[Reviews · NeurIPS 2017]

Reviewer 1



This paper presents an attention-based model for action recognition and human object interaction. The model can be guided by extra supervision or not. It achieves accuracy improvements without increasing the network size and computational cost a lot. Authors provide an extensively empirical and analytical analysis of the attention module, and they further introduce a derivation of bottom-up and top-down attention as low-rank approximations of bilinear pooling methods for fine-grained classification typically. Experiments on three benchmarks have demonstrated the efficacy of the proposed method. One key advantage is to learn the attention map in an unsupervised manner. To make a prediction, the attention maps can provide insight into where the network should look in terms of both bottom-up saliency and top-down attention. This allows it to get rid of detecting the bounding box usually required in hard attention. This is an interesting work, and the idea sounds well motivated. The paper reads well too. One major concern is: The fully-connected weight matrix in second-order pooling is approximated by the product of two vectors with rank-1. Then, how about the information loss compared the original one? Does such loss affect the attention map? Some theoretical analysis and more discussions are expected. Minor issues: Page 3, "network architecture that incorporate(s) this attention module and explore(s) a pose ..." Page 5, "It consist(s) of a stack of modules" Page 8, "... image resized to 450px at(as) input time did"?

Reviewer 2



The paper introduces a simple yet efficient way of incorporating attention modelling in CNN-based action recognition networks. The paper is clearly written, the contribution is interesting and the experimental validation is convincing. Minors comments: Section 2 speaks of "performance" without explicitly stating which evaluation metrics are considered, except for [36] where it is said that mean average precision is considered (btw the acronym mAP is not properly defined here), this should be fixed to be clearer. A few typos were left that need further proofreading.

Reviewer 3



The authors propose to use a low rank approximation of second order pooling features for action recognition. They show results on three compelling vision benchmarks for action recognition and show improvements over the baseline. Pros: (+) The proposed method is simple and can be applied on top of any network architecture (+) Clear improvement over baseline Cons: (-) Unclear novelty over well known second order pooling approaches (-) Fair comparison with other attention based models is missing It is unclear to me how the proposed approach is novel compared to other low rank second order pooling methods, typically used for fine grained recognition (e.g. Lin et al, ICCV2015), semantic segmentation etc. While the authors do a decent job walking us through the linear algebra and interpreting the variables used, the final approach is merely a classifier on second order features. In addition, the authors make design choices that are not clearly justified. For multi class recognition, the authors do not justify the choice for a class agnoistic b but for class-specific a. These experiments should be added in order to prove the effectiveness of the proposed design. The authors claim that they get rid of the box detection step present at other approaches such as R*CNN or Mallya & Lazebnik. However, they do not discuss how their approach would generalize to instance-specific action recognition. An image can contain many people who perform different actions. In this case, the action labels are not assigned at the frame level, but at an instance level. R*CNN was designed for instance-level recognition, thus the necessity of the person box detection step. The authors should discuss how the proposed approach can handle this more generic setup of instance-level labels? This is a limitation of the current approach, rather than a benefit as implied in the Related Work section. The authors fail to provide a fair comparison with competing approaches. The authors use a ResNet101 or an Inception-V2 architecture for their base network. They show improvement over their own non-attentive baseline, using the same architecture. However, the competing approaches, such as R*CNN or Mallya & Lazebnik all use a VGG16 network. This makes comparison with these approaches and the current approach unfair and inconclusive on MPII and HICO dataset. On HMDB, the improvement over the TSN BN-inception method is small while comparisons with other methods is unclear due to the varying underlying base architectures.